# Evaluation and Analysis of Influencing Factors of Ecosystem Service Value Change in Xinjiang under Different Land Use Types

**Yang Wang** *,†, **Remina Shataer** †, **Zhichao Zhang, Hui Zhen and Tingting Xia**

Xinjiang Key Laboratory of Grassland Resources and Ecology, College of Grassland Science, Xinjiang Agricultural University, Urumqi 830052, China; 13079904349@163.com (R.S.); adzhangzhichao@163.com (Z.Z.); zhenhuihuizi@163.com (H.Z.); xiatt1027@163.com (T.X.)
* Correspondence: ktwangyang@163.com; Tel.: +86-13999252425
† These authors contributed equally to this work.

**Abstract:** Based on the data on land-use change in Xinjiang from 1990 to 2020, this study uses a combination of land-use dynamics, the equivalence factor method, the sensitivity index, and a spatial correlation study to quantitatively analyze the spatial and temporal distribution of land-use change and ecosystem service values in the study area from 1990 to 2020. We also use a geographic probe model to explore the driving mechanism of the spatial variation of ecosystem service values in Xinjiang. The following conclusions were drawn: (1) land use in the Xinjiang region from 1990 to 2020 shows a more drastic change, with the main characteristics being an increase in the area of arable land and construction land, and a decrease in the area of forest and grassland, water, and unused land; and (2) with the change in the land-use types, the total value of the ESV in the Xinjiang region from 1990 to 2020 showed an increasing and then decreasing trend, with an average annual contribution value of about $13,730.33 \times 10^8$ yuan and a cumulative loss of about $1741.00 \times 10^8$ yuan in the last 30a. The value of each individual ecosystem service was dominated by functions such as waste treatment and water connotation. Based on the analysis of the geographic probe model, we found that the single factor influence degree of the ESV was HAI > NDVI > precipitation > GDP > temperature > elevation > population density > slope, and the overall ecosystem service value in the Xinjiang region showed a decreasing trend due to the interaction coefficients of natural factors and socio-economic factors.

**Keywords:** land-use change; ecosystem service value; geographic detector model; driving mechanism; Xinjiang region





## 1. Introduction

Ecosystem services provide an essential benefit to humans, and they are classified according to different types of land use by dissecting the variability and distribution of ecosystem service values in total and spatial terms [1,2]. At the early stage of studying ecosystem service values, these two factors are often taken as the main topics of research, and they are conducive to grasping the changes in regional ecosystem services and the differences in ecological levels. Moreover, they provide strong theoretical support for the formulation of ecological and environmental policies and the rational use of land resources [3]. Ecosystem services originate from the protection of the environment and its impact on ecosystems and related processes, and are combined with different driving factors. These factors include natural and human factors such as land use, urbanization, and environmental management [4]. With the in-depth research on the value of ecosystem services, scholars at home and abroad began to study the relevant influencing factors and their interactions [5,6].

The Xinjiang region is located in a mountain–oasis–desert system, which is closely linked to the constituents in the temporal and spatial dimensions, and has a complex material exchange and energy exchange, creating a rich, green ecosphere structure for regional development [7,8]. In the context of complex socio-ecological systems, the increasingly prominent contradiction between ecology and economic development, the greater demand for resources for socio-economic development, and the profound impact of human activities undoubtedly cause negative impacts for sustainable socio-economic development [9]. The ecological environment in arid areas is extremely fragile, and even small changes in the ecosystem may have a huge impact on regional ecological security [10]. Therefore, in recent years, the evaluation of the ecosystem service value, the characteristics of temporal and spatial differentiation, and the analysis of influencing factors have become hot topics in the related research. Meng [11] used the revised ecosystem service value coefficient, based on the typical Tarim River, Shiyang River, and Heihe oasis wetlands in arid oasis regions, and used the benefit transfer method to explore the difference in ecosystem service values among the three areas. The research results showed that, due to the over-exploitation of oasis wetlands, the wetland area has been drastically reduced, the landscape has been destroyed, and the ecological benefits of the wetlands have been lost. However, with the implementation of the national wetland protection policy, the wetland area and ecological benefits in the middle reaches of the Heihe River, the Tarim River Basin, and the middle reaches thereof had significantly improved. Luzhou [12] and other researchers used the arid desert area of Alxa as a typical study area and utilized the InVEST model to evaluate the impact of land-use change in Alxa on the value of regional ecosystem services. They found that human factors are the cause of the rapid decline in the quality of the ecological environment in the area. Therefore, the adoption of an ecological immigration policy could gradually restore and strengthen the service functions of the ecological environment, and thereby promote the sustainable development of the region. Liu [13] took the ecological restoration project area of the Irtysh River Basin as an example, and scientifically implemented the ecological protection project of mountains, rivers, forests, fields, lakes, and grasses. Based on this, the ecological protection area and ecological red line area were reasonably demarcated, and a reasonable ecological compensation system was established. Wang [14] comprehensively analyzed the change rate of ecosystem service functions and the impact of spatial aggregation on ecological functions, and proposed a dynamic reconstruction method based on ecological service priority indicators and spatial enrichment indicators. Three different levels of ecological security models were established. Dade M. C. [15] proposed that the positive synergy and negative trade-off relationships between ecosystem services are affected by driving factors, and a process-based model can be used to achieve the effective management of ecosystem services. At present, the research and analysis of ecosystem service values in the author's country is mainly carried out in consideration of the aspects of natural factors, human factors, and the sustainable development of ecosystem value, as well as multi-scale research on the changes to regional ecological service values [16]. Wang [17] studied the contribution rate of the changes in the ESV in the Xilingol Grassland Reserve, showing that the loss of the ESV in the reserve due to natural climate change was the highest, and proposing that an ecological compensation mechanism for nature reserves should be reasonably established. Chen [18] used multi-scale buffer gradients to quantitatively analyze the evolution of urban landscape patterns and to assess the gains and losses in the ESV, revealing the temporal and spatial evolution characteristics of landscape patterns and the ecosystem service value (ESV) at different buffer scales. Guo [19] conducted an in-depth analysis of the driving factors by constructing a comprehensive evaluation index system for assessing desertification sensitivity and introducing geographic detectors and other methods, using remote sensing technology and GIS spatial analysis technology over a research area in the arid region of northwest China. They showed that, among the driving factors in the study area, soil and climate play a direct role, and vegetation is the most basic factor in changing the sensitivity to desertification.



In summary, the literature on the influencing factors of ecosystem service values has been gradually enriched, from focusing on land-use change in the early stage to the response of the climate to hydrological changes [20], the expansion of the urban scale [21], and social development [22]. However, it cannot be ignored that the breadth and depth of the research remain slightly insufficient [23]. To date, the driving forces of land ecology have attracted the attention of many scholars, mainly using qualitative and correlation analysis methods. The quantitative identification and interaction mechanism of driving forces is crucial for regional habitat protection and the effective formulation of relevant policies [24,25]. The detection model is a new type of spatial statistical method, which has been widely used in the study of ecological engineering protection, the drive towards the expansion of construction land, and food security [26–28]. Few studies have considered the interactions of ecological service values at a large regional scale, according to the detection model.

## 2. Data and Materials

### 2.1. Description of the Study Area

Xinjiang is located in the hinterland of the Eurasian continent, at 73°40′–96°18′ E, 34°25′–48°10′ N, with an altitude ranging from −217–8483 m. It has a temperate continental climate with an average annual temperature of 9.72 °C, an average annual precipitation of about 135.31 mm, and a large difference in temperature between day and night; the temperature difference in most areas is as high as 20 °C. The vertical natural belt spectrum in Xinjiang is obvious, with glaciers, tundra, forests, grasslands, deserts, and other landscapes constituting a unique natural ecosystem and landform unit due to multiple factors such as the geological structure and climate of the arid region. The spatial and temporal distribution of water resources in the region is extremely uneven, the vegetation coverage is low, and the ecosystem is fragile and strongly affected by climate change. In recent years, the water area has been greatly reduced, forests and grasslands have been severely damaged, and the ecological functions of grasslands and forests have been continuously weakened. The total study area is $1.67 \times 10^6$ square kilometers, and it has jurisdiction over four prefecture-level cities, five regions, and five autonomous prefectures. Its ecological location and geographic strategic position are very prominent [29] (Figure 1).

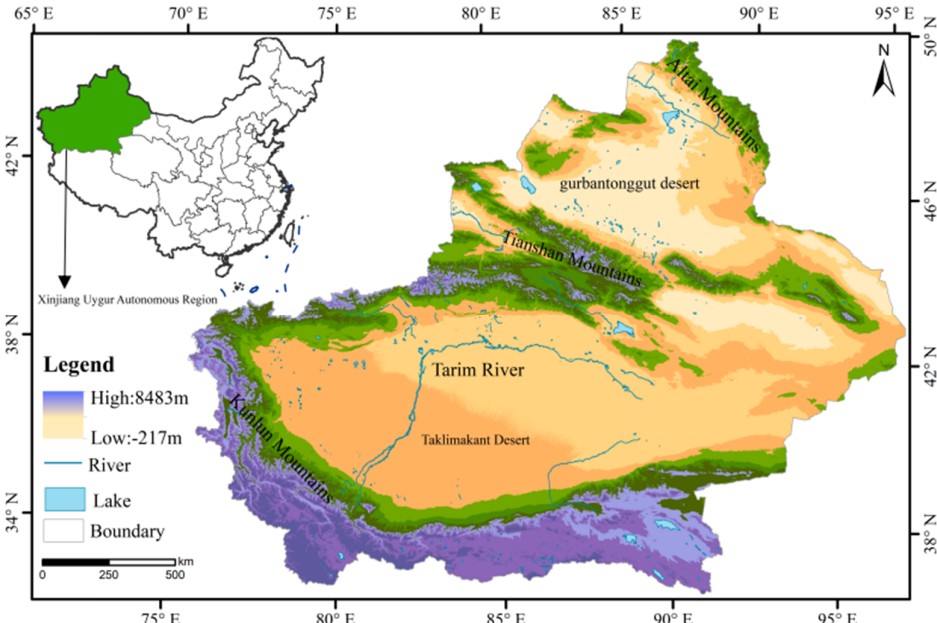

**Figure 1.** Sketch map of the study area.

*2.2. Methods*

Based on the disciplines of geography, ecological economics, and environmental science, this paper uses 3S technology to establish a dynamic land-use model and transfer matrix based on land-use data to analyze the spatial and temporal changes of land use in the Xinjiang region and different ecological zoning units from 1990–2020. In this study, the average price, yield, and sown area of food crops in the administrative units of the Xinjiang Uygur Autonomous Region from 1990 to 2020 were statistically calculated, and the results showed that the ecosystem equivalent factor of the study area was about 1881.82 yuan/hm$^2$, based on which the ecological service value of the Xinjiang region, north of Tianshan Mountain, and south of Tianshan Mountain was assessed. In this paper, with reference to related studies [30], the Xinjiang region was divided into 3422 30 km × 30 km square grid cells with the help of the ArcGIS Fishnet tool to reveal the changes in regional ecological service values and the spatialized display of the intensity of anthropogenic disturbance drivers. The temporal and spatial dimensions were used to explore the evolution trend of regional ecological service values over 30 years. The study on the related driving forces was mainly conducted in regard to natural and socio-economic factors using statistical analysis, correlation analysis, and geographic probes, so as to provide a scientific basis for the construction and sustainable development of ecological civilization in the Xinjiang region.

2.2.1. Data Collection and Processing

(1)  The land-use data were selected from the 30 m spatial resolution land-use data in Xinjiang published by the Resource Science and Data Center of the Chinese Academy of Sciences (https://www.resdc.cn/, accessed on 15 August 2021), and the Xinjiang region was selected for image processing and analysis.

(2)  The NDVI data were obtained from NASA EOS/MODIS data (http://wist.echo.nasa.gov/api, accessed on 22 October 2021). The NDVI data come from the MOD13Q1 product, with a spatial resolution of 250 m and an interval of 16 d. The Xinjiang region involves six rows and columns (h23v04, h23v05, h24v04, h24v05, h25v04, and h25v05) with a total of 2760 remote sensing images.

(3)  The DEM data of the SRTM in the study area with a spatial resolution of 30 m were obtained from the Geospatial Data Cloud (http://www.gscloud.cn, accessed on 22 September 2021).

(4)  The daily precipitation data were derived from the data of 33 meteorological stations including Habahe, Jinghe, Bayinbulak, and Shache, and the temperature data were obtained from the China Meteorological Data Network (http://data.cma.cn/, accessed on 11 January 2022).

(5)  The socio-economic data were obtained from the Xinjiang Statistical Yearbook and the Xinjiang Production and Construction Corps Statistical Yearbook (1990–2020) (Table 1).

**Table 1.** Data Sources.

| Type of Data | Time | Data Attributes | Source |
|---|---|---|---|
| Land-use data | 1990–2020 | Land-Use Status Data | Chinese Academy of Sciences |
| Terrain data | | SRTM 30 m DEM Data | Geospatial Data Cloud |
| Remote sensing data | 2000–2020 (16d) | MODIS Image | NASA |
| Meteorological data | 1990–2020 | Precipitation, Temperature | China Meteorological Data Center |
| | | | Xinjiang Statistical Yearbook |
| Socio-economic data | 1990–2020 | Socio-economic Indicators | Xinjiang Production and Construction Corps Statistical Yearbook |

### 2.2.2. Single Dynamic Degree of Land Use

The land-use dynamic degree is an index used to evaluate the rate and magnitude of changes in different land-use types within a certain time range [31], reflecting the impact of human activities on a single land-use type. Its expression is:

$$K = \frac{U_j - U_i}{U_i} \times \frac{1}{t} \times 100\% \tag{1}$$

where $K$ is the dynamic degree of a certain land-use type, $U_i$ and $U_j$ represent the areas of the land-use type at the beginning and end of the research period, respectively, and $t$ is the research time. The larger the value of $K$, the more obvious the dynamic change of the considered land-use type.

### 2.2.3. Ecosystem Service Value

In this evaluation of the ecosystem service value, the Chinese scholar Xie [32] took part of the results of the Costanza R [33] evaluation model as a reference and revised it, in order to formulate the appropriate Chinese ecosystem service value equivalent according to the actual situation in China. In this formulation, cultivated land corresponds to farmland, forest land corresponds to forest, grassland corresponds to grassland, water corresponds to rivers and lakes, and unused land corresponds to desert [34]. According to the average price of grain, unit yield, planting area, and other indicators in each administrative division of Xinjiang from 1990 to 2020, the equivalent factor of the ecological service value in Xinjiang is 1881.82 yuan/hm$^2$, which was used to calculate the ecological service value coefficient per unit area of the study area (Table 2). In this paper, the service values for the ecosystems in the study area were calculated according to the service value coefficient and the area of each category in the north of the Tianshan Mountains from 1990 to 2020. The calculation formula is as follows:

$$ESV = \sum Aa \times VC_a \; ESV = \sum Aa \times VC_{ba} \tag{2}$$

where $ESV$ is the ecosystem service value, $A_a$ is the area of the $a^{\text{th}}$ land-use type in the study area, $VC_a$ represents the ecological service value coefficient of the $a^{\text{th}}$ land-use type, and $VC_{ba}$ represents the $b^{\text{th}}$ ecosystem service value of the $a^{\text{th}}$ land-use type.

**Table 2.** Ecosystem service value coefficients in Xinjiang (yuan/hm$^2$).

| Ecosystem Service Function | Land-Use Type | | | | | |
|---|---|---|---|---|---|---|
| | Cultivated Land | Forest Land | Grassland | Water | Construction Land | Unused Land |
| Gas regulation | 940.91 | 6586.37 | 1505.46 | 0.00 | 0.00 | 0.00 |
| Climate regulation | 1674.82 | 5080.92 | 1693.64 | 865.64 | 0.00 | 0.00 |
| Water conservation | 1129.09 | 6021.83 | 1505.46 | 38,351.50 | 0.00 | 56.45 |
| Soil formation and protection | 2747.46 | 7339.10 | 3669.55 | 18.82 | 0.00 | 37.64 |
| Waste disposal | 3086.19 | 2465.18 | 2465.18 | 34,211.50 | 0.00 | 18.82 |
| Biodiversity conservation | 1336.09 | 6134.74 | 2051.18 | 4685.73 | 0.00 | 639.82 |
| Food production | 1881.82 | 188.18 | 564.55 | 188.18 | 0.00 | 18.82 |
| Raw material production | 188.18 | 4892.73 | 94.09 | 18.82 | 0.00 | 0.00 |
| Entertainment culture | 18.82 | 2408.73 | 75.27 | 8167.10 | 82.60 | 18.82 |
| Total | 13,003.38 | 41,117.78 | 13,624.38 | 86,507.29 | 82.60 | 790.36 |

### 2.2.4. Sensitivity Analysis of the Ecosystem Service Value

In this study, we use the ecosystem service sensitivity index (*CS*) to clarify the degree of influence of the value coefficient on the value of ecosystem services at different time scales [35]. Its calculation formula is:

$$CS = \left| \frac{\frac{(ESV_j - ESV_i)}{ESV_i}}{\frac{VC_{jk} - VC_{ik}}{VC_{ik}}} \right| \tag{3}$$

where $CS$ is the sensitivity coefficient of the ecosystem value coefficient, and $i$ and $j$ represent the initial ecosystem service value and the value after adjustment using the ecological value coefficient, respectively. If $CS > 1$, ESV is elastic to $VC$, and its accuracy and reliability are low. To the contrary, if $CS < 1$, ESV is inelastic to $VC$, and the research results are credible.

### 2.2.5. Inverse Distance Weighting Method

Based on the basic assumption of the first law of geography, we use a linear weight combination that affects a series of spatial sampling points within the range of the interpolation variable to determine the eigenvalues to be interpolated, where the weight of the influence is inversely proportional to the distance [36]. This is calculated as follows:

$$z = \frac{\sum_{i=1}^{n} \frac{z_i}{(d_i)^p}}{\sum_{i=1}^{n} \frac{1}{(d_i)^p}}, d_i = \sqrt{(x - x_i)^2 + (y - y_i)^2} \tag{4}$$

where $z$ is the predicted value at the estimated point, $Z_i$ is the $i^{th}$ ($i = 1, ..., n$) sample value, $d_i$ is the distance between the estimated point and the interpolation point, and $P$ represents the power of the distance. If the selection criterion is that the power value with the lowest mean absolute error is regarded as the best power value, the higher the power value, the smoother the interpolation effect.

### 2.2.6. Analysis Method for the Ecosystem Service Value's Human Impact Index

The human activity index (HAI) can reflect the impact of human activities on land-use and landscape composition in Xinjiang. Correlations of regional ecosystems were analyzed [37,38]. The formula is as follows:

$$HAI = \sum_{i=1}^{n} \frac{A_i P_i}{TA} \tag{5}$$

where $HAI$ is the comprehensive index of human influence, $A$ is the total area of the $i^{th}$ land type, $P$ is the human-influence intensity parameter reflected by the $i^{th}$ land type, $TA$ is the total land area in the evaluation unit, and $n$ is the number of types of land. In this paper, based on existing research [39], and combined with the characteristics of regional land type attributes, the parameter $P$ is assigned using the assignment method.

### 2.2.7. Geodetector Analysis

Geodetector analysis is used to detect the spatial differentiation characteristics of elements, and includes four types of detectors [40,41]. In this study, factor detectors and interactive detectors were used to identify the dominant factors affecting the spatial differentiation of ESV in Xinjiang, as well as their interactions. The formula is as follows:

$$q = 1 - \frac{1}{N\partial^2} \sum_{h=1}^{L} N_h \partial_h^2 \tag{6}$$

where $h$ is the stratification/partition of the variable $Y$ or factor $F$, $N$ and $N_h$ are the number of units in the overall and $h$ areas, respectively, and $\partial$ and $\partial_h$ are the variance of the Y values in the overall and $h$ areas, respectively.

## 3. Results and Analysis

### 3.1. Analysis of Land-Use Change and Dynamic Attitude in Xinjiang

The interannual change of land-use types in Xinjiang from 1990 to 2020 (Table 2) and the distribution map of land-use/-cover types (Figure 2) show that unused land was the land type with the highest coverage distribution in the study area, accounting for 61.05% on average. The grassland area showed a U-shaped dynamic change with a decreasing and then increasing trend, but it was not significant. Cultivated land was the most dominant

land-use type in the study area, and had the largest increase in area, whereas the water area showed a decreasing trend. The construction land area was the smallest, accounting for 0.25–0.53% of the total area.

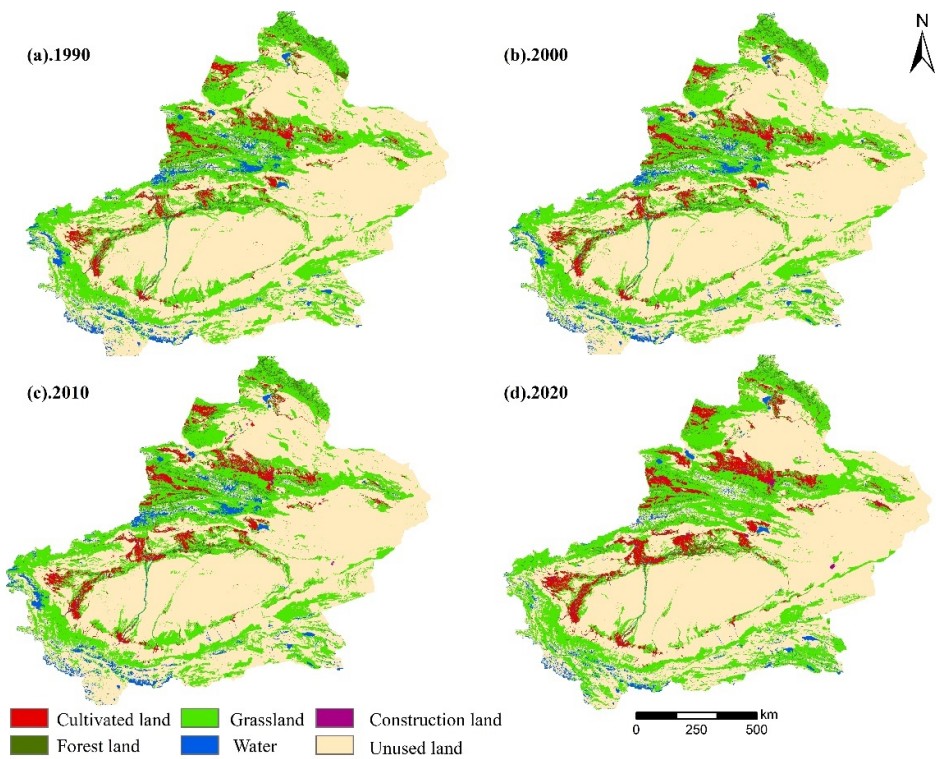

**Figure 2.** Land-use status in Xinjiang from 1990 to 2020.

From 1990 to 2020, the land-use types in the study area showed different changes in different periods, which were mainly manifested as an increase in cultivated land and construction land area, and a decrease in forest and grassland, water, and unused land area. During the study period, the increase in cultivated land area was the most obvious, with a cumulative increase of $335.28 \times 10^4$ hm$^2$ over the past 30 years (Table 3). The growth rate was the fastest from 2010 to 2020, with a dynamic attitude of 3.10. This change was mainly caused by the conversion of grassland ($286.31 \times 10^4$ hm$^2$) and unused land ($104.43 \times 10^4$ hm$^2$), and the cultivated land area increased, to a certain extent, due to the reclamation and protection of wasteland. The total growth rate of construction land was $86.37 \times 10^4$ hm$^2$ during the study period, which showed the fastest growth rate from 2010 to 2020, and a dynamic attitude of 7.37, mainly from cultivated land ($29.30 \times 10^4$ hm$^2$), unused land ($20.08 \times 10^4$ hm$^2$), and grassland ($16.16 \times 10^4$ hm$^2$). Unused land was the largest land-use type in the study area; its area peaked at $1117.98 \times 10^4$ hm$^2$ in 2000 and decreased to $1010.03 \times 10^4$ hm$^2$ in 2020 (with a decrease of about $107.98 \times 10^4$ hm$^2$). During this time, the decrease rate was the fastest from 2000 to 2010, and it was mainly converted into grassland ($1240.95 \times 10^4$ hm$^2$) and cultivated land ($104.43 \times 10^4$ hm$^2$). This may be related to the implementation of ecological policy protection by the state to make efficient use of land in order to meet the needs of high-quality urban development. During the study period, the water area fluctuated and decreased, with a positive dynamic increase from 1990 to 2010 and a sharp decrease of $176.47 \times 10^4$ hm$^2$ from 2010 to 2020. This area was mainly converted to unused land ($190.42 \times 10^4$ hm$^2$) and grassland ($84.86 \times 10^4$ hm$^2$). The forest and grassland area decreased by $108.98 \times 10^4$ hm$^2$ and $43.31 \times 10^4$ hm$^2$ from 1990 to 2020, respectively. Woodland decreased at the fastest rate from 2010 to 2020, whereas grassland increased slightly, with a dynamic attitude of −2.67 and 0.20, respectively. The forest and grassland areas mainly converted to cultivated land and unused land. This change reflects that, with the influence of economic and social development and policy

factors, forest and grassland areas are shrinking due to excessive farmland reclamation, grazing, and deforestation.

**Table 3.** Land-use transfer matrix for Xinjiang from 1990 to 2020 ($\times 10^4$ hm$^2$).

| Land-Use Type | Cultivated Land | Forest Land | Grassland | Water | Construction Land | Unused Land | Total | Transfer Out |
|---|---|---|---|---|---|---|---|---|
| Cultivated land | 458.69 | 11.99 | 55.28 | 4.04 | 29.30 | 6.79 | 566.09 | 107.40 |
| Forest land | 31.38 | 135.58 | 186.22 | 5.09 | 1.75 | 23.44 | 383.47 | 352.08 |
| Grassland | 286.31 | 108.55 | 3252.99 | 53.42 | 16.16 | 1148.71 | 4866.14 | 4579.84 |
| Waters | 4.44 | 2.98 | 84.86 | 211.26 | 0.89 | 190.42 | 494.86 | 490.42 |
| Construction land | 16.12 | 1.01 | 2.53 | 0.26 | 18.19 | 2.26 | 40.37 | 24.24 |
| Unused land | 104.43 | 14.38 | 1240.95 | 74.80 | 20.08 | 8551.56 | 10,006.19 | 9901.76 |
| Total | 901.37 | 274.49 | 4822.83 | 348.86 | 86.37 | 9923.18 | 16,357.10 | |
| Transfer in | 442.68 | 262.50 | 4767.56 | 344.82 | 57.07 | 9916.39 | | |

### 3.2. Temporal and Spatial Variation of the Ecosystem Service Value

3.2.1. Temporal Variation Characteristics of the Ecosystem Service Value

Based on the ecosystem service value coefficients for Xinjiang, we estimated the ecosystem service value of each region from 1990 to 2020 (Table 4). The temporal and spatial loss characteristics of ecosystem service values are closely related to the geographical environment. Considering topography, climate, and other related factors, we conducted a regional analysis on southern and northern Xinjiang. From 1990 to 2020, the ecosystem service values in Xinjiang generally decreased. In the first 20 years, the ESV in the study area showed a gentle upward trend, with a cumulative increase of 104.51 $\times$ 10$^8$ yuan. From 2010 to 2020, the ESV decreased sharply, with a loss of about 1372.35 $\times$ 10$^8$ yuan, among which the value of water areas lost the most, accounting for 72.8% of the total loss. The natural climate of Xinjiang is arid, and the lack of precipitation supply and the over-exploitation of water resources are the main factors aggravating the reduction in the ESV.

**Table 4.** Changes in the ESV of various land-use types in Xinjiang from 1990 to 2020 (10$^8$ yuan).

| Partition Unit | Land-Use Type | 1990 | | 2000 | | 2010 | | 2020 | |
|---|---|---|---|---|---|---|---|---|---|
| | | ESV | % | ESV | % | ESV | % | ESV | % |
| Xinjiang | Cultivated Land | 737.95 | 5.23 | 771.6 | 5.43 | 896.94 | 6.31 | 1144.04 | 9.24 |
| | Forest Land | 1578.28 | 11.18 | 1569.43 | 11.05 | 1542.967 | 10.85 | 1101.78 | 8.9 |
| | Grassland | 6663.25 | 47.2 | 6534.12 | 45.99 | 6456.51 | 45.4 | 6414.83 | 51.84 |
| | Water | 4344.82 | 30.78 | 4538.16 | 31.93 | 4531.76 | 31.87 | 2948.25 | 23.82 |
| | Construction Land | 0.33 | 0 | 0.36 | 0 | 0.41 | 0 | 0.69 | 0.01 |
| | Unused Land | 791.66 | 5.61 | 795.24 | 5.6 | 792.21 | 5.57 | 765.71 | 6.19 |
| | Sub-total | 14,116.29 | 100 | 14,208.91 | 100 | 14,220.797 | 100 | 12,375.30 | 100 |
| Northern Xinjiang | Cultivated Land | 417.66 | 7.99 | 422.83 | 8.04 | 483.61 | 9.14 | 617.21 | 12.42 |
| | Forest Land | 1030.28 | 19.71 | 994.62 | 18.92 | 994.16 | 18.79 | 601.23 | 12.10 |
| | Grassland | 2683.36 | 51.33 | 2675.31 | 50.90 | 2634.62 | 49.79 | 2879.46 | 57.96 |
| | Water | 833.66 | 15.95 | 901.59 | 17.15 | 918.53 | 17.36 | 622.08 | 12.52 |
| | Construction Land | 0.2 | 0.00 | 0.24 | 0.00 | 0.27 | 0.01 | 0.47 | 0.01 |
| | Unused Land | 262.02 | 5.01 | 261.88 | 4.98 | 260.13 | 4.92 | 247.94 | 4.99 |
| | Sub-total | 5227.18 | 100 | 5256.46 | 100 | 5291.31 | 100 | 4812.02 | 100 |
| Southern Xinjiang | Cultivated Land | 318.01 | 3.64 | 349.89 | 3.99 | 412.23 | 4.71 | 544.58 | 7.20 |
| | Forest Land | 553.45 | 6.34 | 581.82 | 6.63 | 555.21 | 6.34 | 517.75 | 6.85 |
| | Grassland | 3930.1 | 45.02 | 3807.99 | 43.38 | 3773.46 | 43.10 | 3621.95 | 47.89 |
| | Water | 3402.85 | 38.98 | 3510.64 | 39.99 | 3486.94 | 39.83 | 2352.25 | 31.10 |
| | Construction Land | 0.13 | 0.00 | 0.12 | 0.00 | 0.14 | 0.00 | 0.24 | 0.00 |
| | Unused Land | 524.3 | 6.01 | 528.04 | 6.02 | 526.78 | 6.02 | 526.51 | 6.96 |
| | Total | 8728.84 | 100 | 8778.5 | 100 | 8754.76 | 100 | 7563.28 | 100 |

From 1990 to 2020, the ESV trends of northern Xinjiang and southern Xinjiang were basically the same, with a maximum contribution value of $5291.31 \times 10^8$ yuan in 2010 and a minimum contribution value of $4968.39 \times 10^8$ yuan in 2020. Therefore, the ESV loss from 2010 to 2020 was the highest at about $-1845.50 \times 10^8$ yuan. Grassland contributed the most to the total ecosystem service value, accounting for 45.4–51.84%. In addition, the ESV contributions of cultivated land and construction land also showed obvious upward trends, with a large inter-annual change rate of ESV. The ESV contribution of other land types decreased to different degrees, with the greatest decrease being for water areas, followed by forest land. From 1990 to 2020, the ESV in southern Xinjiang showed a trend of fluctuating decline, with slow growth in the first 10 years and a sharp decline of 13.61% in the last 10 years. From the perspective of the change of different land types, the ESV of water, forest, and grassland in southern Xinjiang decreased, with the loss of water areas being the largest, with a total decrease of $1050.60 \times 10^8$ yuan (or $-30.87\%$) during the study period. The ESVs of the other areas showed an increasing trend. It can be seen that the sharp decline in the water area was the fundamental reason for the decline in the ESV value in southern Xinjiang from 1990 to 2020.

Ecosystem services are processes based on natural ecosystem functions that directly or indirectly serve human beings, which can be divided into four categories, according to their different functional values: regulation, support, supply, and culture [42]. According to the first-level classification of the ESV, the proportions of the ESV contribution rates of different regional units in Xinjiang from 1990 to 2020 were as follows: regulation service > support service > supply service > cultural service (Table 5). The ecological regulation service value in northern Xinjiang was the most outstanding, accounting for 60.11% of the total value. The contribution rate of support services in southern Xinjiang was 32.88%. The proportions of supply and cultural services were not significant (4.61% and 3.96%, respectively).

**Table 5.** Changes in the value of individual ecosystem services in various units in Xinjiang ($10^8$ yuan).

| Type 1 | Type 2 | Xinjiang | | Northern Xinjiang | | Southern Xinjiang | |
|---|---|---|---|---|---|---|---|
| | | ESV | % | ESV | % | ESV | % |
| Regulation Service | Gas regulation | 991.43 | 8.01% | 459.14 | 7.24% | 532.37 | 7.04% |
| | Climate regulation | 1032.25 | 8.34% | 462.95 | 8.10% | 569.34 | 7.53% |
| | Water conservation | 2387.46 | 19.29% | 753.31 | 21.73% | 1633.88 | 21.60% |
| | Waste disposal | 2553.75 | 20.64% | 854.1 | 23.04% | 1699.38 | 22.47% |
| | Sub-total | 6964.88 | 56.28% | 2529.5 | 60.11% | 4434.97 | 58.64% |
| Support service | Soil formation and protection | 2256.35 | 18.23% | 1025.21 | 15.78% | 1231.29 | 16.28% |
| | Biodiversity conservation | 2076.12 | 16.78% | 821.04 | 15.39% | 1255.28 | 16.60% |
| | Sub-total | 4332.47 | 35.01% | 1846.25 | 31.17% | 2486.57 | 32.88% |
| Provision of services | Food production | 472.18 | 3.82% | 218.65 | 2.94% | 253.59 | 3.35% |
| | Raw material production | 196.83 | 1.59% | 100.50 | 1.72% | 96.80 | 1.28% |
| | Sub-total | 669.42 | 5.41% | 319.14 | 4.66% | 350.39 | 4.55% |
| Cultural service | Entertainment culture | 408.52 | 3.30% | 117.13 | 4.06% | 291.35 | 3.85% |
| | Total | 12,375.30 | 100% | 4812.02 | 100% | 7563.28 | 100% |

Individual ecosystem service values can further reflect the composition of the regional ecosystem structure. From 1990 to 2020, regulation services in Xinjiang increased first and then decreased, and the ESV of water conservation and waste treatment decreased significantly, with decreases of $626.78 \times 10^8$ yuan and $658.78 \times 10^8$ yuan, respectively. Support services include soil formation and conservation and the maintenance of biodiversity, and the single ESVs tend to increase slowly. Soil formation and protection (20.63%) and food production (4.4%) were the main sources of ESV contribution in northern Xinjiang. In southern Xinjiang, the ESV of food production showed a trend of slow increase, while other ESV values showed declining trends. This phenomenon indicates that, although the unit ecosystem area in southern Xinjiang is large, the value of ESV had decreased from 1990 to 2020, mainly caused by natural and human factors in the region. The trend of

regulating services increasing first and then decreasing kept a high correlation with the changes in water area and unused land area. The lack of water resources and desert area in southern Xinjiang resulted in a negative development trend of the mutual transformation of ecosystem functions in the region. In northern Xinjiang, the vegetation coverage is good, the fluctuation of forest and grassland is reduced, and the support services are greatly affected. This may be due to the local overloading of grazing and logging, which lead to decreases in forest and grassland areas. Supply services and cultural services accounted for a small proportion of the ecosystem service value in the study area. Among them, the food production function of supply services was mainly affected by cultivated land and construction land, which was consistent with the change trend of cultivated land and construction land area, showing a continuous increasing trend. The cultural service function increased slowly from 1990 to 2000 and decreased sharply from 2010 to 2020. The change of the cultural service function in this region was synchronous with that of the unused land and water area.

In conclusion, the value of ecosystem services in Xinjiang showed a fluctuating trend of decline from 1990 to 2020, consistent with the change trend of the ESV in northern Xinjiang. Over the past 30 years, ESVs in both northern and southern Xinjiang showed decreasing trends, but the value loss in southern Xinjiang ($-1165.55 \times 10^8$ yuan) was much higher than that in northern Xinjiang ($-415.17 \times 10^8$ yuan), mainly due to the large area of desert distributed in the middle of the basin in this region, and the trend of expansion. Moreover, the area of oasis surrounding the desert is shrinking. In addition, grassland resources are abundant in northern Xinjiang, and forest and grassland areas have significant impacts on ecosystem service values. Therefore, the protection of forests and grasslands is crucial to the ecological conservation of the study area.

### 3.2.2. Change Characteristics of the Spatial Dimension of the Ecosystem Service Value

In order to further analyze the spatial and temporal distribution characteristics of ecosystem service values in Xinjiang, we created a Fishnet tool based on the ArcGIS 10.2 platform. We used this tool to calculate the ecosystem service value within each grid scale. In order to display the change of the land ecosystem service value in the study area from 1990 to 2020 more directly, the natural break point classification method in ArcGIS was used to divide the ecosystem service values in the Xinjiang region from low to high in six grades, with I being the lowest and VI the highest. In this way, we can explore the changes in ecosystem service values by displaying them spatially (Figure 3).

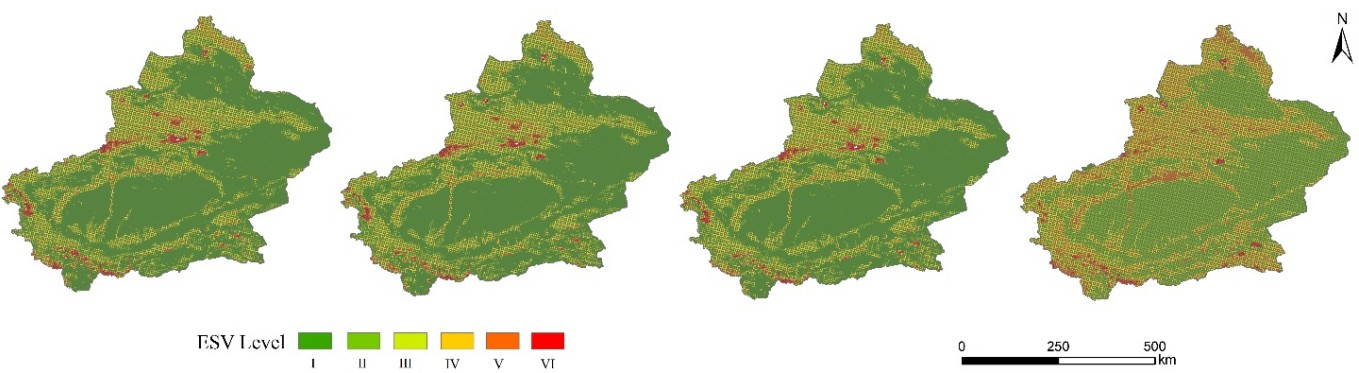

**Figure 3.** Spatial distribution characteristics of the ESV in Xinjiang from 1990 to 2020.

From 1990 to 2020, the spatial and temporal changes in the ecosystem service value grades in Xinjiang showed that the basic pattern in the study area was relatively stable, showing a distribution trend of "high in the north and southwest, low in the middle and southeast". This feature may be highly consistent with the typically mountainous oasis–desert transition zone in Xinjiang. Areas with a high ESV grade are mainly distributed in mountain areas, and the landforms are mainly mountainous and hilly, with rich natural forest and grassland resources and a good spatial combination. The ESV is mainly

distributed in oasis areas, and the land type is mainly cultivated land and construction land. The low-grade ESV area was largely distributed in desert units, among which unused land is a relatively large land-use type. Compared with the spatial distribution changes of the ESV in 1990 and 2020, the area of high-grade ESV increased annually, while that of low-grade ESV did not change significantly, but accounted for a large proportion of the total area. The original class I area had expanded to various degrees, mainly due to the rapid development of the city, resulting in the transformation of cultivated land and unused land into construction land. In the middle part of the study area, the sporadic grade VI changes to grade V, due to the relative increase in construction land area in the center, resulting in a relative decrease in forest and grassland area and a decline in the ESV.

Over the past 30 years, Xinjiang's urbanization process has accelerated with the rapid development of the country's economy. Considering the gains and losses of land-use and ESV, it is not difficult to find that some areas have changed to a certain extent. In the north and southwest of the study area, the distribution characteristics are high-grade, and the ESV is high. Affected by the policies for returning grazing land to grassland and farmland protection, the areas of arable land and grassland increased steadily, while the unused land area decreased. The distribution characteristics in the "central and southeast" of the study area are low-grade. This is due to the Taklamakan Desert, the largest desert in China, being distributed in the central region, which has a large desert area and a low ESV.

### 3.2.3. Sensitivity Analysis

According to the sensitivity calculation formula, the *VC* of each land-use type in the study area was adjusted up and down by 50%, respectively, in order to obtain the sensitivity index, *CS*, for the study area in 1990, 2000, 2010, and 2020 (Figure 4). The results indicated that the *CS* values of different land-use types in the four periods were less than 1, where the maximum *CS* value was obtained for grassland (0.5184); that is, when the *VC* of grassland increased by 1%, the total value increased by 0.5184%. The minimum value was construction land, with a *CS* value of 0.0000. From 1990 to 2020, the *CS* value of cultivated land increased, while the *CS* values of forest land and water area decreased, which was due to the large-scale reclamation of forest land and grazing in the study area, resulting in a sharp decrease in forest land, the continuous expansion of cultivated land, the increased demand for agricultural water irrigation, and a decreased water area. The *CS* value for grassland showed a fluctuating upward trend, reaching a maximum value in 2020. This indicates that grassland contributed the most to its ecosystem services, and this was related to the large increase in area in the eastern Tianshan Mountains and Bayanbulak Grassland in Xinjiang during 2010–2020 due to the strengthening of grassland protection under the guidance of national policies. *CS* sensitivity analysis showed that the sensitivity index of each region in the study period was less than 1. Therefore, the ESV in Xinjiang is inelastic to *VC*, the result is reliable, and the value coefficients are suitable for calculating the ecosystem service values in Xinjiang.

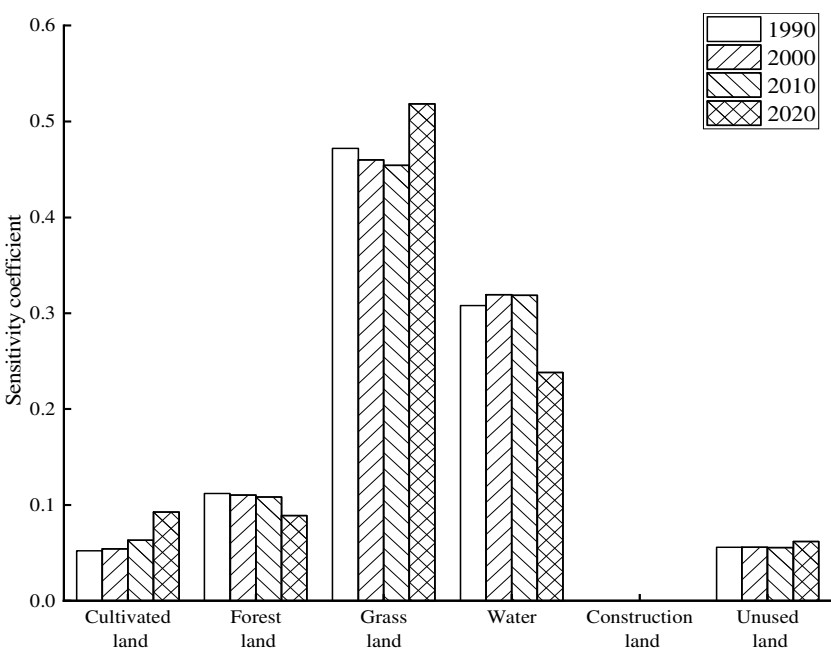

**Figure 4.** Sensitivity coefficients of the ESV for various land-use types in Xinjiang from 1990 to 2020.

*3.3. Study on the Driving Mechanism of the Ecosystem Service Value*

3.3.1. Impact of Climate Factors on the Ecosystem Service Value

Our research demonstrated that the ecological service values in Xinjiang had changed from 1990 to 2020, in a manner immeasurably related to natural and human factors. In terms of natural factors, we used the correlation spatial analysis method to further analyze the impacts of climate, terrain, and water system distribution on the ecosystem service values. From 1990 to 2020, the temporal and spatial distribution of ESV in the study area showed a negative correlation with the average annual temperature (Figure 5). Due to the high altitude, low temperature, and abundant precipitation in northern Xinjiang, there were abundant forest and grass resources, and the ESV showed a high trend; meanwhile, southern Xinjiang has low altitude, high temperature, drought and little rain, sparse grassland vegetation, and low ESV. The Tarim Basin in southern Xinjiang is the driest area in China. The center of the basin is the hinterland of the Taklamakan. The extreme maximum temperature is 41–45 °C, and the evaporation capacity in the basin is strong and greater than the precipitation. Turpan Basin in northern Xinjiang is a typical graben basin and is the basin with the lowest altitude in China at about −154.31 m, which leads to a dry climate, little rain, windy sand, and an extremely arid landscape pattern. During the study period, the temporal and spatial distribution of ESV in Xinjiang was generally positively correlated with the average annual precipitation. The precipitation distribution is uneven, and the areas with high precipitation are distributed from west to east along the Tianshan Mountains. Northern Xinjiang is significantly higher than southern Xinjiang and has more mountainous areas than basin areas. The increase in precipitation promotes the contribution of the ESV, to a certain extent. From the single service value perspective, the contribution ratio of regulation service and supply service in northern Xinjiang is high. Water conservation and the regulation of the supply can ensure stable grain production.

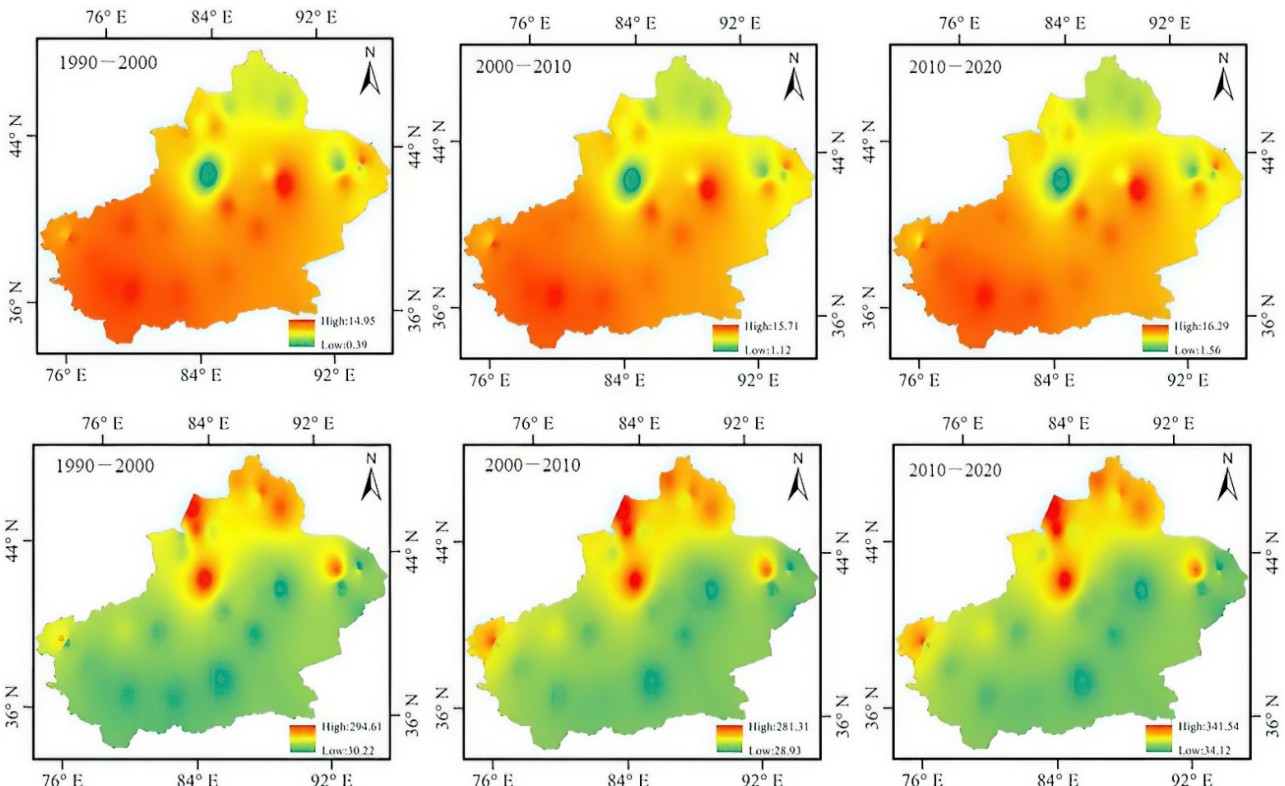

**Figure 5.** Distribution characteristics of the spatial variation of temperature and precipitation in Xinjiang from 1990 to 2020.

3.3.2. Impact of Natural Vegetation Differentiation on the Ecosystem Service Value

The special geographical environment of Xinjiang has led to a significantly diverse vegetation distribution. During the five periods from 2000 to 2020, the NDVI of natural vegetation in the study area changed significantly, where the changes in vegetation coverage showed the spatial distribution characteristics of "the oasis is high in the desert, while northern Xinjiang is high and southern Xinjiang is low" (Figure 6). According to the change in vegetation cover, the spatial distribution characteristics of NDVI were divided into five grades, namely $0 < \text{NDVI} \leq 0.2$, $0.2 < \text{NDVI} \leq 0.4$, $0.4 < \text{NDVI} \leq 0.6$, $0.6 < \text{NDVI} \leq 0.8$, and $0.8 < \text{NDVI} \leq 1.0$. The areas with high NDVI values were mostly distributed in the mountains in the north and south of the study area and showed a pattern of marginal river growth; the areas with low NDVI values were the vast areas in the middle of the study area, where there are large bare desert areas in the centers of the basins. The zonal distribution of desert vegetation in the arid region of northwest China affects the spatial transfer of ESV resources. During the study period, the high-value area ($0.8 < \text{NDVI} \leq 1.0$) showed a very significant growth trend, from $156.92 \times 10^4$ hm$^2$ in 2000 to $473.89 \times 10^4$ hm$^2$ in 2020, for an increase of 201.99%. The $0.6 < \text{NDVI} \leq 0.8$ area was large and distributed in the Altai Mountains and the Tianshan Mountains in the northern part of Xinjiang while, in the southern part, it surrounds the basin in the form of a belt. This area showed a trend of continuous expansion. The $0.4 < \text{NDVI} \leq 0.6$ area was mainly distributed in the middle-altitude area, where the water and heat conditions are conducive to the growth of vegetation, but it is easily disturbed by human activities, and the vegetation cover was destroyed, resulting in an area of $1096.52 \times 10^4$ hm$^2$ in 2000, and an area of $1094.65 \times 10^4$ hm$^2$ in 2020. The area showed a dynamic change of fluctuating decrease. The $0.2 < \text{NDVI} \leq 0.4$ areas were mostly distributed in oases and along riverbanks. Due to the influence of human irrigation needs and the growth of natural vegetation on abandoned cultivated land, this type of area also showed growth. The $0 < \text{NDVI} \leq 0.2$ areas were mostly distributed in the southern and central parts of the study area, located in the cross belt of oasis and desert, where the growth conditions required by vegetation are poor, and the available

water is low. Due to the comprehensive impact of the natural environment and human land-use, this area showed a continuous decreasing trend from 2005, having decreased from $12{,}300.96 \times 10^4$ hm$^2$ in 2000 to $11{,}706.16 \times 10^4$ hm$^2$ in 2020, for a total reduction of $594.80 \times 10^4$ hm$^2$, decreasing at a rate of $29.74 \times 10^4$ hm$^2$/a. Forest and grassland are representative high-value ESV areas in Xinjiang, which had significant contributions to individual values. The mountain grassland in the northern part of the study area showed the most abundant vegetation.

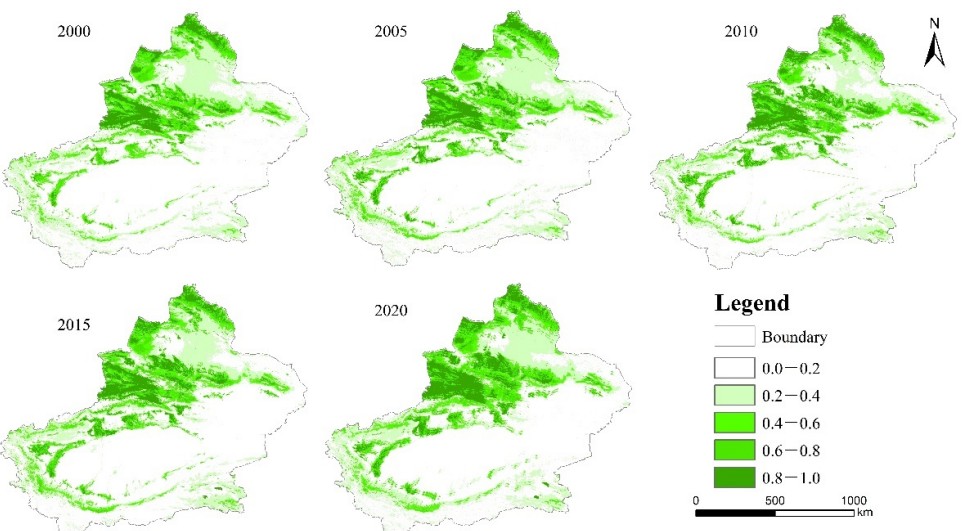

**Figure 6.** Distribution characteristics of the spatial variation of NDVI in Xinjiang from 2000 to 2020.

### 3.3.3. The Impact of Human Activities on Ecosystem Service Values

To assess the impact of human activities on ecosystem service values, we used the ArcGIS 10.2 platform to reveal the driving strength of human disturbance in Xinjiang from 1990 to 2020. The spatialized display is shown in Figure 7. In order to better reflect the intensity level of human disturbance in the study area from 1990 to 2020, the natural break point method in ArcGIS was used to classify the intensity of the ecosystem service value in Xinjiang. From low to high, it was divided into five categories: low impact ($0 < \text{HAI} \leq 0.2$); medium–low impact ($0.2 < \text{HAI} \leq 0.4$); medium impact ($0.4 < \text{HAI} \leq 0.6$); medium–high impact ($0.6 < \text{HAI} \leq 0.8$); and high impact ($\text{HAI} > 0.8$). These were used to reveal the anthropocentric interference intensity in the Xinjiang region from 1990–2020 [43].

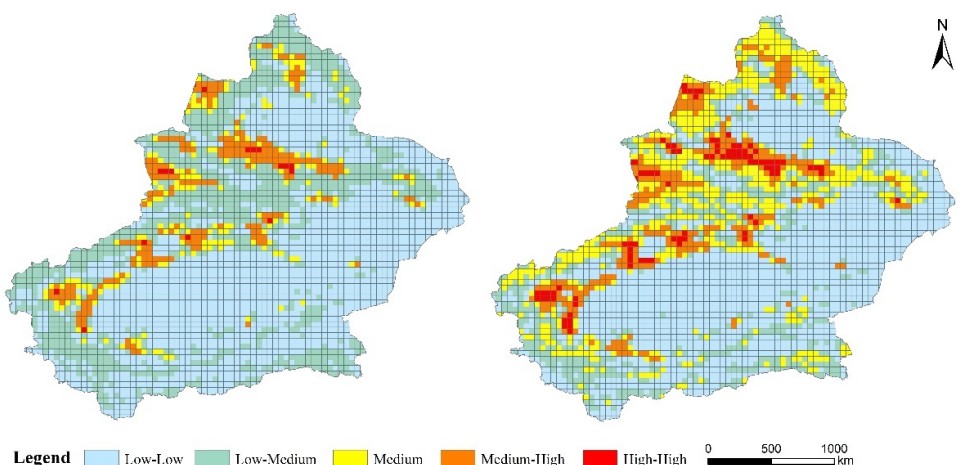

**Figure 7.** Distribution map of the comprehensive intensity of human interference in Xinjiang from 1990–2020.

From the figure, it can be seen that from 1990 to 2020, the basic pattern of the spatial distribution of anthropocentric disturbance intensity is obvious, showing a distribution trend of "high in the north and west, and low in the middle and south". Low impact intensity is dominant and is mainly distributed in the desert hinterland, where the inherently harsh ecological environment is not suitable for the occurrence of human activities. Although the range of low impact intensity on the regional scale is wide, the proportion gradually decreased over time. The low impact intensity was concentrated along the edge of the oases and high mountain areas on the periphery of the desert. The landform types in these areas are mainly mountainous and hilly, and the intensity of human activities is low. Compared with the spatial distribution changes of the influence intensity in 1990 and 2020, the medium influence intensity range continued to expand and spread, and the proportions in the northern, central, and southern parts of the study area increased to varying degrees. The medium–high influence intensity areas showed initial sporadic spots, which were transformed into patches with a certain shape and aggregation. These areas were mainly distributed in the oases, and the land types were mainly cultivated land and construction land; therefore, human activities in these areas are frequent. The high-impact intensity area increased with urbanization. Its scope increased slightly, but the acceleration of urbanization will inevitably affect the development of the regional ESV.

*3.4. Geographical Detection Analysis of the Spatial Differentiation of the Ecosystem Service Value*
3.4.1. Impact Factor Detection and Analysis

The spatial differentiation of ecosystem service values in Xinjiang is driven by multiple comprehensive factors. We selected eight driving factors that affect the value of regional ecosystem services at the two levels of natural environment and social environment. It can be seen from Figure 8 that the explanatory power of these factors with respect to the ESV in Xinjiang can be arranged as follows: HAI > NDVI > precipitation > average GDP > air temperature > altitude > population density > slope. The contribution rate of HAI was the highest (at 59.3%), indicating that the ecological environment in this area is under multiple influences of human socio-economic activities, of which human factors are the most important. The contribution of the normalized vegetation index (NDVI) was lower than that of the HAI (43.5%), where the NDVI was the key factor affecting the spatial differentiation of the ESV in the study area. The contributions of altitude, precipitation, and average GDP ranged from 3.3–15%, making them relatively important factors affecting the spatial differentiation of the ESV in the study area. The contributions of population density and slope were small, and they had little effect on the spatial differentiation of the ESV in the study area. Natural and socio-economic factors in the study area explain the magnitude of the impact of spatial differences in the ESV. Compared with natural environment factors, socio-economic factors had greater impacts on the spatial differentiation of the regional ESV, especially the anthropocentric impact index among the socio-economic drivers. The HAI is the key factor affecting the spatial differentiation of the ESV, closely followed by the normalized vegetation index (NDVI), an important influencing factor affecting the change of the spatial differentiation pattern of the ESV in the study area. Specifically, socio-economic factors were the primary factors affecting the ESV, while certain natural economic factors had a greater impact on it. Therefore, the ESV is influenced by both socio-economic and natural factors. Under the background of global changes and intensified human activities, the ecological environment in Xinjiang is constantly weakening and the contradiction between man and land is increasing, which seriously hinders its ecological and cultural development.

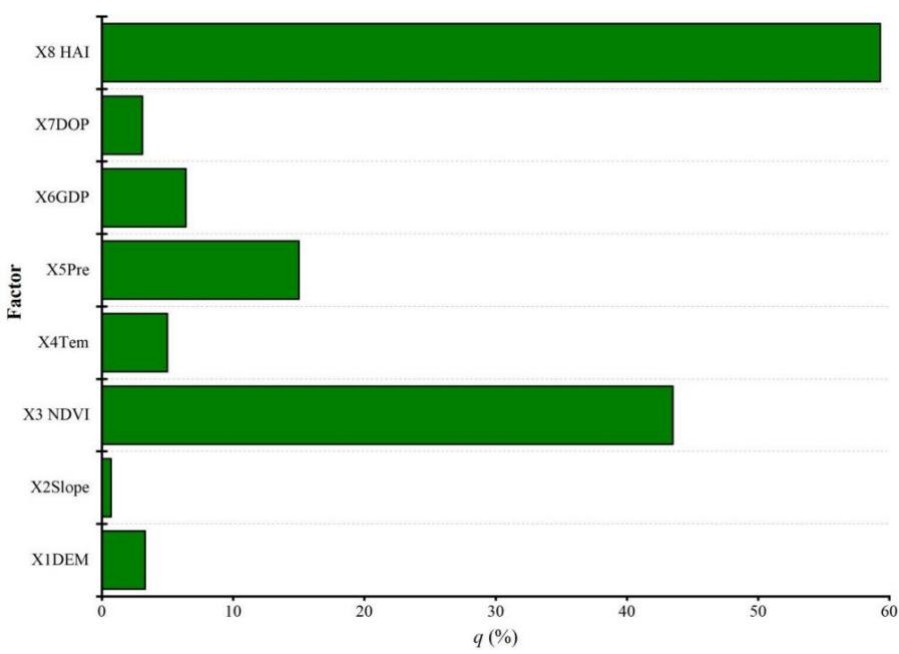

**Figure 8.** Factor Detection of the Spatial Differentiation of ESVs.

3.4.2. Interaction Influence Analysis

The detector model can be used to detect the interaction between two factors. The influence of a given factor on the ESV is not independent, as interactions between factors may occur, where the interaction mode is mainly non-linear enhancement and two-factor enhancement. According to the research results shown in Table 6, the HAI∩NDVI interaction had the strongest impact on the spatial differentiation of the ESV, with the highest $q$-value of factor interaction detection (0.665) and an explanatory power of nearly 67%. Other detected interactions included HAI∩elevation ($q = 0.643$), NDVI∩elevation ($q = 0.639$), HAI∩precipitation ($q = 0.631$), HAI∩average GDP ($q = 0.618$), HAI∩average GDP ($q = 0.618$), HAI∩air temperature ($q = 0.611$), HAI∩slope ($q = 0.610$), HAI∩population density ($q = 0.602$), and NDVI∩slope ($q = 0.577$). Although the $q$-values of the interaction types of the other factors were all less than 50%, the effect of two factors on the spatial differentiation of the ESV was always higher than that of the single factors.

**Table 6.** Factor Interaction Detection for the Spatial Differentiation of ESVs.

| Factor | X1 DEM | X2 Slope | X3 NDVI | X4 Tem | X5 Pre | X6 GDP | X7 DOP | X8 HAI |
|---|---|---|---|---|---|---|---|---|
| X1 DEM | 0.033 | | | | | | | |
| X2 Slope | 0.047 | 0.007 | | | | | | |
| X3 NDVI | 0.639 | 0.577 | 0.435 | | | | | |
| X4 Tem | 0.084 | 0.062 | 0.481 | 0.050 | | | | |
| X5 Pre | 0.201 | 0.179 | 0.485 | 0.193 | 0.150 | | | |
| X6 GDP | 0.116 | 0.081 | 0.480 | 0.130 | 0.201 | 0.064 | | |
| X7 DOP | 0.061 | 0.045 | 0.448 | 0.071 | 0.164 | 0.096 | 0.031 | |
| X8 HAI | 0.643 | 0.610 | 0.665 | 0.611 | 0.631 | 0.618 | 0.602 | 0.593 |

In the study area, the interactions between human activities and other factors were much stronger than the effects within a single factor. This was mainly due to the complex mechanism of human activities: the spatial distribution of the ESV under regional geographic and climatic conditions is strongly affected by external human factors, which greatly increase the impact of the differences in spatial distribution. In addition, the interaction effects of NDVI and other natural factors also increased significantly, indicating that the normalized vegetation index under Xinjiang's hydrothermal and terrain conditions is an important determinant of spatial distribution. At the same time, the synergy of natural

factors–such as annual precipitation and NDVI–and socio-economic factors, as well as the interactions between socio-economic factors, enhanced the impact of regional ecosystem service values. Therefore, in order to optimize the ecosystem structure and reasonably assess and avoid ecosystem risks, the interactions of multiple influencing factors need to be considered.

## 4. Discussion

At present, Xinjiang is located in the core area of the "One Belt and One Road" initiative, and its ecological problems are one of the biggest obstacles to the "clear waters and green mountains" ecological restoration process in China [44]. In recent years, with the growth of the population, indiscriminate deforestation and grazing have intensified the demand for land resources, resulting in forest land degradation and land desertification in the region, leading to an increasingly fragile environment [45,46]. In terms of exploring the driving mechanisms of the ecosystem service value in Xinjiang, our approach was consistent with that of Li et al. [47], who selected the important factors affecting the spatial differentiation of the ESV in the study area based on natural and humanistic factors. The contribution of HAI to driving factors affecting the ecosystem service value was high in the study area, indicating that complex human economic and social activities cause disturbances to ecosystems to a certain extent, in contrast to the results from previous studies considering different spatial scales. For example, Han [48] found that the agricultural population, total industrial output value, and tourism income were the main factors influencing the spatio-temporal differences of the ESV in eastern Sichuan. Wang [49] researched the temporal and spatial variation rules and driving factors of the ecosystem service value in Leshan city and showed that natural factors were more decisive than social economic factors; however, social economic factors had an increasingly prominent influence. Jiang [50] conducted a study at the scale of the Shiyang River Basin and showed that meteorological and hydrological conditions were important factors leading to regional habitat change. Zhou [51] found that ESV in their study area was significantly negatively correlated with the urbanization rate and significantly positively correlated with the gross forestry product, which were the main driving factors. In addition, the ESV was coupled with social and economic development, and the urbanization rate had a negative effect, while the increase in gross forestry product increased the ESV. The driving factors of ecosystem services include internal factors and external factors. Previous studies have mostly focused on the mechanisms of external factors, such as social and economic development, the promotion of urbanization, and the dynamic spatial evolution of land. However, the roles of spatial interactions have not been fully discussed.

The value of ecological services in Xinjiang has undergone certain changes, which have an immeasurable relationship with both natural and human factors [52]. Existing studies have mostly aimed to estimate the impact of social and economic activities on ecosystems, where land-use change accounts for a certain proportion of regional ecosystems. Due to the expansion of cultivated land and construction land, large-scale forests, grasslands, and water areas have been reduced, which, in turn, affects the total ecosystem service value. The dynamic change of land use is closely related to the value of ecological services. Therefore, how to reduce the transformation of higher-value land resources under the premise of ensuring social and economic development is a difficult, yet key issue in the study of the ecosystem service value [53]. In addition, during the study period, unused land in the south of the Tianshan Mountains in Xinjiang accounted for a large proportion, but still showed a small expansion trend. If no measures are taken to control this, the extensional expansion of the Taklamakan Desert may lead to the degradation of the total value of ecological services in the study area. The negative effect will continually increase, restricting the development process of the ecological environment in the study area [54]. Most of the rivers in the arid region of the northwest originate from high mountainous areas. The mountain areas are rich in precipitation and are the formation area of surface runoff; this water system is relatively developed. It is worth noting that water resources

run through the mountain–oasis–desert complex ecosystem. With the increase in global temperature and the severe impact of human activities on the current state of the water resources in the study area, the water area has decreased to a large extent, the lake area has shrunk, and desertification has intensified. As such, the single value of the ESV has been indirectly degraded and the habitat condition is severe [55,56]. Therefore, this study provides a relevant theoretical basis for the sustainable development of regional ecosystems. We should focus on strengthening the management and control of water resources, protect land with water, and rely on the material and economic prosperity of oases to promote the improvement and development of mountain and desert ecosystems to achieve socio-economic and ecological benefits [57]. We must further practice the dynamic balance of supply and demand between ecological and social systems.

The prominent contradiction between ecology, economy, and development affects the structure and function of the entire ecosystem of arid areas. Considering this, multiple control and management strategies can be adopted, and the land-use development mode that is suitable for the regional natural conditions and social and economic development level should be selected to avoid unreasonable land use and development. Otherwise, the synergistic effect of strong anthropogenic land-use disturbance and natural and socio-economic factors is expected to increase the pressure on the regional ecosystem [58,59]. According to the above discussion, in terms of research on the ecosystem service value, in this paper, we used the revised unit area value equivalent factor method for estimation, and mainly explored the temporal and spatial variation characteristics and driving factors of the ESV in the study region. The service value was generally low, and a complete regionally applicable evaluation system has not been formed; therefore, the data acquisition degree in all aspects was not sufficient. It is hoped that more comprehensive data can be obtained in the future, in order to reduce the errors in the research and improve the accuracy of the research results. In addition, with updated high-precision satellite remote sensing data and the release of socio-economic data, our follow-up research will use urban agglomerations at the regional scale to explore the urban development and migration trends, driven by factors such as policy, population, and economy. With the in-depth exploration of relevant theories and methods, focusing on the ecological service mechanism in Xinjiang, future work may uncover more detailed relationships [60,61].

## 5. Conclusions

Based on land-use data from 1990, 2000, 2010, and 2020, combined with the actual situation of the study area, we used the equivalent factor correction method and spatial autocorrelation method to quantitatively analyze the spatio-temporal loss and driving mechanism of ecosystem service values in Xinjiang. Aiming to provide practical implications for regional ecological service value assessment, our main conclusions are as follows:

(1) From 1990 to 2020, land use in Xinjiang showed relatively dramatic changes. With the year 2010 as the boundary, the area of arable land and construction land continued to increase, while the area of forest and grassland, watershed, and unused land fluctuated and decreased. Land-use conversions in Xinjiang were frequent during the study period, showing an increase in the transfer of arable land and construction land, and a significant transfer of forest and grassland, watershed, and unused land. In terms of different zoning units, the increase in grassland area is more obvious in the northern Tianshan region, while there was a successive expansion of unused land area and a reduction in the water area in the southern Tianshan region.

(2) The ESV value of the Xinjiang region from 1990 to 2020 showed a general decreasing trend, from $14{,}116.31 \times 10^8$ yuan in 1990 to $12{,}375.30 \times 10^8$ yuan in 2020. Among them, the ESV of grassland and watershed are high, and the increase in the construction land is the most significant. The ESV classification of Xinjiang and each ecological unit is as follows: regulating services > supporting services > supplying services > cultural services. The spatial distribution pattern of the ESV is obvious, with the distribution

pattern of "high in the north and southwest, and low in the center and southeast". The value of ecosystem services shows obvious spatial correlation and aggregation.

(3) The influence degree of single driving factors on the ESV in Xinjiang from 1990 to 2020 can be ranked as follows: HAI > NDVI > precipitation > average GDP > air temperature > elevation > population density > slope. The contribution of HAI was the highest (at 59.3%) and, thus, human influence was the core driving factor in the spatial variation of the ESV in the study area.

**Author Contributions:** Y.W. conceived the study design and implemented the field research, R.S. and Z.Z. collected and analyzed the field data; H.Z. and T.X. applied statistics, mathematics, or other forms of technology to analyze or research data. R.S. wrote the paper with the help of Y.W. All authors have read and agreed to the published version of the manuscript.

**Funding:** This research was funded by the Natural Science Foundation of Xinjiang Uygur Autonomous Region (2021D01E02); National Natural Science Foundation of China: "Dynamic evolution of desertification process based on the changes of bare desert boundary" (Grant No: 41661015).

**Institutional Review Board Statement:** Not applicable.

**Informed Consent Statement:** Not applicable.

**Data Availability Statement:** Not applicable.

**Conflicts of Interest:** The authors declare no conflict of interest.

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
