# Peer review of "Evaluation and Analysis of Influencing Factors of Ecosystem Service Value Change in Xinjiang under Different Land Use Types"

_water, doi:10.3390/w14091424_

Round 1

Reviewer 1 Report

REVISION COMMENTS

The article approaches a very interesting and timely topic. The article scope fits within this journal aim and follows a clear structure. It is highly valuable the multiscale approach to unveil the influencing factors in Ecosystem Services value change in relation with different land use types. The selected case of study, the Xingjian Region in China, is undoubtedly complex and permits to conduct crosscutting perspectives.

Some comments are given in order to improve and clarify different aspects of the revised manuscript.

Abstract

As long as the abstract can determine whether or not other researchers are going to be interested in the research work, I suggest to shorten the abstract extension. Authors should decide if is more convenient to develop an informative approach –purpose, method, scope, results– outlining the main information without going into the details; or, alternatively, to provide a descriptive abstract trying to engage the audience. In both cases, the goal is to inform, so results can be the biggest section of the abstract, but as succinct summary highlighting the relevant points.

I recommend to avoid stating the results as obvious as it undermines the relevance of the research carried out. If some of the results reinforce the initial hypothesis, it should be explained providing background information in reference to other studies, or advising interpretation of the findings. But, in any case, I would avoid making these approaches in the abstract section.

Introduction

The introduction approaches a theoretical framework but some excessively wordy sentences and long paragraphs prevent from an agile reading. It is difficult to grasp the statement of the importance of the topic addressed for:  society, the discipline, or, the identification of controversy within the field of study, among other aspects.

I suggest to additionally introduce the reasons for selecting the region of Xingjian. Some interesting features about this region are given in the first paragraph of the discussion section (lines 553-561), but few is explained about why this study case is selected, and how resulting outcomes could be of the interest for an international researcher.  

Data and Materials

I suggest to increase accuracy when describing features. For example:

(line 123) “… difference between day and night is large.” I would include the range variation of temperatures with figures or percentages.

(line 125) “and so on”  is a vague expression and accuracy is expected.

(line 125) “constitute a unique natural ecosystem” Why? Which are the features? It is of the interest of the research to provide evidence-based details.

(line 131) “1.6649 million square kilometers”, I suggest to round the figure 1.67x106 square kilometers

In the “Methods” subsection it would be desirable to provide a starting paragraph giving the reasons to, for example, have adopted a specific method, the criteria for selection, the innovative approach due to the combination of indicators, the different penetration and granularity of each source,…

Regarding the subsection 2.2.3 "Ecosystem service value", there is a central question in relation with the equivalent factor yuan/hm2 (line 172).

In the extract, there's only a single representative value, with no additional information from the year it is based in. It should be specified if this is an average value from the year range provided (1990-2020). Most importantly, if that is not the case, further specifications should be made for a clearer visualization of the value's evolution throughout the time period, and how it has adjusted to economic transitions. 

Has this factor been considered the same value without taking into account the economic changes during the last 30 years? Should we suppose that there is no variation in the service value coefficient related to timeframe or is it only referred to 2020? This is a core issue that should be clarified. It could be also interesting to include the equivalence in Euros or US dollars together with the Yuan currency, to provide a more international perspective.

Additionally, the concept of “unused land” remains confusing because, maybe is a “non productive land”, but, is a broad term which should be disambiguated to understand why it emerges as a specific category apart from natural land, or grassland, etc.

In the same subsection, Table 2 (line 181) shows the ecosystem service value coefficients organized per functions, but it is difficult to appreciate the criteria. For example, in the "ecosystem service function" column, there seems to be no strategy of ranking by importance, priority or any other criteria that would help to better understand the data and its impact.

Results and Analysis

Line 223. “The grassland area showed a u-shaped dynamic change, but it wasn’t significant”. This type of statements should provide specific evidence based on the results obtained.

From line 229 to 256 several figures are given, and it is difficult to follow the long-winded explanation, furthermore, it is referred as table 2 (line 233) and it should be referred to table 3. Some of the figures, such as the cumulative increase of cultivated area 335.28x104 hm2, are not reflected in any of the tables.

In the 3.2.1. subsection is introduced a subdivision of the study area into Northern and Southern subregions. It would be highly recommended to advance these steps in the research methodology, so that it would help to follow the process more easily.

The Results and Analysis section includes conclusions in some of the subsections (for example lines 322-331, lines 518-521), and it is followed by a Discussion section and finally a Conclusion section. I recommend restricting all the information related to discussing the findings with more elaborate commentaries to the Discussion section. In the case that the authors prefer to combine both, I would suggest eliminating the Discussion section and elaborate a unified “Results and Discussion” section.

In line 361 it is stated “over the past 30 years, with the Belt an Road Initiative,…” which has not been previously mentioned, so it is difficult to understand the implications and repercussions for the assessment of Ecosystem Service value.

In the subsection 3.3.3. “the impact of Human activities…” (line 463) states that “the study area was divided into 3422 30 kmx30 km grids”. It is necessary to explain why 30x30 is the selected grid, which was the scientific reasons to finally selected this pattern. What is more, this explanation should be included into the methodology section, not in the results one.

Discussion

I suggest to better highlight the contribution of this research. In my opinion one of the key points is reflected in the second paragraph (lines 588-591)

Results

Apart from summarizing the main research findings, it would be necessary to suggest more general implications for the field of knowledge, showing how this methodology could be of the interest of different regional areas to assess ecosystem services value change, setting out recommendations for practice or policy where appropriate.

English language

English language is not my native language but I have detected some excessively long sentences, and too wordy language, mainly in the introduction section ( for example lines 46-53). Additionally, excesive long paragraphs in different sections make it difficult to have a fluent reading.

References

Finally, regarding the references, it must be highlighted that there are a 22.4% of references (13/58) from the same journal Acta Ecologica Sinica. My suggestion is that it should be justified whether the journal is a highly specialised source for the subject matter of the article or there are other acceptable reasons.

Additionally, I would like to highlight that it has not been possible to locate and cross-check the reference number 35 “Chen, F.; Ge, X.P.; Chen, G.; et al . Spatial Different Analysis of Landscape Change and Human Impact in Urban Fringe. Scientific 755. Geographica Sinica. 2001(03): 210-216”

Author Response

Thank you for your comments concerning our manuscript entitled “Evaluation and analysis of influencing factors of ecosystem service value change in Xinjiang under different land use types”(Water-1680546). Those comments are all valuable and very helpful for revising and improving our paper, as well as the important guiding significance to our researches. We have studied comments carefully and have made correction which we hope meet with approval. Revised portion are marked in red, blue and yellow in the paper. The main corrections in the paper and the responds to the comments are in the attachments.

Reviewer 2 Report

Aim of the work is the evaluation and analysis of influencing factors of ecosystem service value change in Xinjiang under different land use types from 1990 to 2020. Some suggestions are provided in order to improve the work:

  • Abstract
    • Summarize the results section, it takes too much time to be read and can be recued with the main essential results
    • Add the main innovative contribution of the work
  • Introduction
    • it is important to say that the ecosystem services provide essential benefit to humans and their classification according to different land use. Some useful references can be: Morano, P., Guarini, M. R., Sica, F., & Anelli, D. (2021, September). Ecosystem Services and Land Take. A Composite Indicator for the Assessment of Sustainable Urban Projects. In International Conference on Computational Science and Its Applications (pp. 210-225). Springer, Cham and Metzger, M., Rounsevell, M. D. A., Acosta-Michlik, L., Leemans, R., & Schröter, D. (2006). The vulnerability of ecosystem services to land use change. Agriculture, ecosystems & environment114(1), 69-85.
  • Data and methods
    • Add between lines 120-133 the reference to Figure 1 and a brief description of it
    • if possible, add a map of the land use type of the area
    • the period of time for the remot sensing data used doesn't overlap with the others (e.g. 2000-2020 vs 1990-2020), can you explain this?
    • How the U of Eq. (1) parameters were calculated? What do you means with "the areas of the land-use type", the extention of it in kmq? Explain this.
  • Results and Analysis 
    • Can you add, if possible, a map of the Ecosystem Services delta assessed?

Author Response

(The authors gave the same response as above.)

Round 2

Reviewer 1 Report

Firstly, I would like to thank the authors for the effort of providing a point by point answer to all the issues arised in the previous review report. 

The manuscript has been clearly improved, but there are still some minor issues that I recommend to check again:

  1. Abstract: I suggest to reconsider the way it has been structuctured, and look for a more catchy and concise way to present the significant content of your research.
  2. Table 3 remains difficult to comprehend, it only shows the "transfer out" but no conclusion can be made since there are no explicit results for "transfer in". Please reconsider clarifying these results.
  3. On a final note, the use of English should be edited overall for a better readability.

Author Response

Thank you for your letter and for the reviewers’ comments concerning our manuscript entitled “Evaluation and analysis of influencing factors of ecosystem service value change in Xinjiang under different land use types”(Water-1680546). Those comments are all valuable and very helpful for revising and improving our paper, as well as the important guiding significance to our researches. 

Reviewer 2 Report

The efforts made by the Authors are apprecciated, but some suggestions are still not applied, therefore, please, provide for it:

  • Abstract
    • Summarize the results section, it takes too much time to be read and can be recued with the main essential results
    • Add the main innovative contribution of the work
  • Introduction
    • it is important to say that the ecosystem services provide essential benefit to humans and their classification according to different land use. Some useful references can be: Morano, P., Guarini, M. R., Sica, F., & Anelli, D. (2021, September). Ecosystem Services and Land Take. A Composite Indicator for the Assessment of Sustainable Urban Projects. In International Conference on Computational Science and Its Applications (pp. 210-225). Springer, Cham and Metzger, M., Rounsevell, M. D. A., Acosta-Michlik, L., Leemans, R., & Schröter, D. (2006). The vulnerability of ecosystem services to land use change. Agriculture, ecosystems & environment114(1), 69-85.
  • Data and methods
    • the period of time for the remot sensing data used doesn't overlap with the others (e.g. 2000-2020 vs 1990-2020), can you explain this?

Author Response

(The authors gave the same response as above.)
